# Modelling to infer the role of animals in *gambiense* human African trypanosomiasis transmission and elimination in the DRC

**Ronald E. Crump**[1,2]☯*, **Ching-I Huang**[1,2]☯, **Simon E. F. Spencer**[1,3], **Paul E. Brown**[1,2], **Chansy Shampa**[4], **Erick Mwamba Miaka**[4], **Kat S. Rock**[1,2]

**1** Zeeman Institute for Systems Biology and Infectious Disease Epidemiology Research, The University of Warwick, Coventry, United Kingdom, **2** Mathematics Institute, The University of Warwick, Coventry, United Kingdom, **3** The Department of Statistics, The University of Warwick, Coventry, United Kingdom, **4** Programme National de Lutte contre la Trypanosomiase Humaine Africaine (PNLTHA), Kinshasa, Democratic Republic of the Congo

☯ These authors contributed equally to this work.
* r.e.crump@warwick.ac.uk

**Data Availability Statement:** The gambiense human African trypanosomiasis (HAT) data were obtained from the WHO HAT Atlas and are subject

## Abstract

*Gambiense* human African trypanosomiasis (gHAT) has been targeted for elimination of transmission (EoT) to humans by 2030. Whilst this ambitious goal is rapidly approaching, there remain fundamental questions about the presence of non-human animal transmission cycles and their potential role in slowing progress towards, or even preventing, EoT. In this study we focus on the country with the most gHAT disease burden, the Democratic Republic of Congo (DRC), and use mathematical modelling to assess whether animals may contribute to transmission in specific regions, and if so, how their presence could impact the likelihood and timing of EoT.

By fitting two model variants—one with, and one without animal transmission—to the human case data from 2000–2016 we estimate model parameters for 158 endemic health zones of the DRC. We evaluate the statistical support for each model variant in each health zone and infer the contribution of animals to overall transmission and how this could impact predicted time to EoT.

We conclude that there are 24/158 health zones where there is substantial to decisive statistical support for some animal transmission. However—even in these regions—we estimate that animals would be extremely unlikely to maintain transmission on their own. Animal transmission could hamper progress towards EoT in some settings, with projections under continuing interventions indicating that the number of health zones expected to achieve EoT by 2030 reduces from 68/158 to 61/158 if animal transmission is included in the model. With supplementary vector control (at a modest 60% tsetse reduction) added to medical screening and treatment interventions, the predicted number of health zones meeting the goal increases to 147/158 for the model including animal transmission. This is due to the impact of vector reduction on transmission to and from all hosts.

to a data sharing agreement. Interested parties should apply to the WHO (contact J.R. Franco, francoj@who.int) in order to gain access to these data.

**Funding:** This work was supported by the Bill and Melinda Gates Foundation ( www.gatesfoundation. org) through the Human African Trypanosomiasis Modelling and Economic Predictions for Policy (HAT MEPP) project [OPP1177824 and INV-005121] (C.H., R.E.C., P.B., S.E.F.S and K.S.R.) and through the NTD Modelling Consortium [OPP1184344] (K.S.R. and S.E.F.S.). The funders had no role in study design, data collection and analysis, decision to publish, or preparation of the manuscript.

**Competing interests:** The authors have declared that no competing interests exist.

## Author summary

Elimination of African sleeping sickness by 2030 is an ambitious goal, not least because of the unclear role that animals might play in transmission. We use mathematical models, fitted to case data from the DRC to assess and quantify the contribution of animals to the human case burden.

We found that 24 of 158 geographic regions included in this study had statistical support for animal transmission, although it appears extremely unlikely that animals could maintain transmission on their own. Animals could, however, delay elimination; using our model without animal transmission we predicted that 68 regions are expected to achieve elimination by 2030, reducing to 61 with animal transmission. If vector control to reduce fly populations (which transmit the disease to and from hosts) is used in addition to medical interventions, then 147 regions are predicted to reach elimination by 2030 even with animal transmission.

## Introduction

Some infections which cause disease in humans can also be transmitted by non-human animal hosts, increasing transmission opportunities for the pathogen, and potentially hindering control of disease in humans or posing a threat to elimination or eradication. The lack of an animal reservoir is one of the numerous criteria listed as a requirement for a disease to be eradicable [1]. Guinea worm has become notorious for the surprises that can emerge as eradication is approached—despite the huge successes in reducing case reporting from 892,055 in 1989 to under 100 for the first time in 2015, case reduction subsequently stagnated [2] and the parasite has been identified in dogs [3] leading to speculation that this could impede eradication.

The mere presence of human infective parasites in animal populations is not sufficient to warrant immediate concern as this does not preclude the possibility that animals act as "dead-end" hosts, receiving infection themselves whilst not contributing to onward transmission. Even if onward transmission is possible, this alone does not constitute a reservoir; Haydon et al. [4] provide a detailed description of what could be considered a *maintenance* reservoir, specifying that such a population (or collection of populations) must be able to maintain infection on their own, even if infection were (temporarily) eliminated in humans. Of course, even some minor contribution from non-maintenance hosts could slow down progress to achieve an elimination or eradication goal, however control of infection in the human population should eventually enable complete removal of the infection from all human and non-human populations.

All is not lost for the Guinea worm programme [5], although undoubtedly the intensification in eradication efforts so close to zero have required a shift in how the programme must think about control, with uncertainty as to whether the goal can be met and a large price tag. Guinea worm is not alone in this uncertainty—another neglected tropical disease, the *gambiense* form of human African trypanosomiasis (gHAT), is an infection targeted for global elimination of transmission by 2030 [6], yet it remains unclear whether potential animal transmission could prevent this goal from being achieved [7]. gHAT is a vector-borne infection, historically assumed by the infectious disease community to be primarily transmitted to and from humans by blood-feeding tsetse [8]. However, its close relative, *rhodesiense* HAT, is a known zoonosis with most infection occurring in animals [6].

There are a considerable number of reports of *Trypanosoma brucei gambiense* parasites in both wildlife and domestic livestock—Büscher et al. [7] provide a concise recent summary and, in addition, a recent study reported *T. b. gambiense* infections in domestic animals in Chad with 1.2–4.5% of those sampled testing positive in the three main foci [9]. Furthermore, experimental evidence of transmission of *T. b. gambiense* from animals back to humans, through the tsetse vector, has been documented [10], demonstrating that it is possible for animal-tsetse-human transmission cycles to exist. What these data are not telling us directly is whether this transmission pathway is occurring frequently, and to what extent it could be negatively impacting the concerted intervention efforts being made to control gHAT disease in humans.

Over recent decades, control methods for gHAT have primarily focussed on medical interventions in humans; either by diagnosis and treatment of symptomatic individuals at fixed health facilities (passive screening, PS), or through mass screening of at-risk villages (active screening, AS) [11]. There are also a range of methods available to target the tsetse vectors (traps, insecticidal targets, and ground or aerial spraying), although these have not been deployed at scale in many gHAT-endemic regions; in particular the Democratic Republic of Congo (DRC), which has the highest gHAT burden globally, currently only has large-scale tsetse control operations in a handful of its 189 gHAT-endemic health zones due to costs, other resource constraints and logistics over such a large geographical area [12]. Medical interventions, sometimes coupled with vector control, have resulted in a huge decline in transmission since 1998 with many countries achieving low or zero case reporting [6, 13]. The overall success to date certainly could lead us to have an optimistic outlook, yet the history of eradication programmes teaches us that some of the challenges we may face might not become readily apparent until we are very close to zero human cases.

The World Health Organization (WHO) have suggested the assessment of the role of animals as a key priority for gHAT modelling [11, 14]. Modelling studies conducted to date have been performed in a few distinct geographical locations [15–19], but have not tried to assess the questions surrounding the role of animal transmission across larger geographic regions. In one of these studies, prevalence data from humans and various wild and domestic animal species from a focus in Cameroon was used at a single point in time, and found support for the presence of animal reservoirs but did not conclude with certainty whether gHAT could be maintained solely by transmission in animals [15]. Modelling studies in specific foci in the DRC and Chad fitted a gHAT model to longitudinal human case data, but not to animal data, and found similar statistical evidence for model variants including animal transmission compared to those with only human-tsetse transmission. Despite this inconclusive result the model fits were used to conclude that, if transmission from animals is occurring, the animals would not be able to maintain infection without humans, meaning they would not constitute a maintenance reservoir [16, 18, 19]. A final study, that focused on predictions for a focus in Guinea, did not try to quantify whether animal transmission was likely based on case data, but did conclude that, if there were some animal transmission, the likelihood of disease reemergence after stopping interventions would be high. This echoes related findings from a theoretical modelling study of gHAT which suggested that additional interventions, such as vector control, could be needed to curtail transmission if animals were able to acquire and transmit infection [20].

In the present study we seek to build upon this previous work, using human case data from longer time series and with much larger geographic coverage to answer three primary questions: (1) What evidence is there for animal transmission based on longitudinal human case data from across different health zones of the DRC? (2) Are animals maintenance reservoirs? (3) Based on current trends, what will projections to 2030 and beyond look like and what role

does animal transmission play in this (if any)? This third question in particular has important repercussions for planning policy and understanding the route to and sustainability of the 2030 EoT goal.

## Materials and methods

### Data

This study makes use of longitudinal human gHAT data for the DRC from the WHO HAT Atlas spanning the period 2000–2016 [21, 22]. The data contain information on the number of people screened each year in AS campaigns by location, as well as those identified as cases through either AS or PS.

Due to the required diagnostic and treatment algorithm during the data period, initially seropositive cases found in screening with the Card Agglutination Test for Trypanosomes (CATT) or a rapid diagnostic test (RDT) were subsequently confirmed using parasitological techniques before undergoing lumbar puncture to establish whether the infection had progressed to the blood-brain barrier. Finding either trypanosomes or elevated white blood cell (WBC) count ($> 5$ WBC/$\mu$l) denotes a late stage or "stage 2" infection, meaning that patients had to be treated with nifurtimox-eflornithine combination therapy (NECT), as opposed to pentamidine which was used for patients in stage 1. In the WHO HAT Atlas data for the DRC, staging information is typically not available prior to 2015, however this information is mostly present for 2015–16 and has previously proven to be informative when fitting mathematical models to gHAT data [23, 24]. Where available, staging information was included in the fitting, see Fig 1 and S1 Text.

For the purposes of this study we are interested in the health-zone-level aggregated data. Health zones are administrative units of approximately 175 000 people. We therefore take the same approach as utilised in our previous study to extract the data onto a recent shapefile for the DRC [24] before aggregating by year, screening type (active or passive) and staging (stage 1, 2 or unknown) within health zones.

In Crump et al. (2021), where the model without animal transmission was fitted, eight parameters were estimated in the adaptive Markov chain Monte Carlo (MCMC) fitting procedure in all health zones, and an additional four parameters were fitted [24] in health zones where an improvement to PS was modelled (the former provinces of Bandundu and Bas Congo). For the model with animal transmission there are two additional parameters related to transmission to/from animal hosts that we chose to estimate within the model fitting. As a result, we restricted the data to health zones containing at least 13 aggregated data points, that is either an AS coverage of more than 20 people for the year or a non-zero number of cases detected by PS, this being three more aggregated data points than the 10 required in the earlier study with a slightly less complex model [24]. Consequently we find a total of 158 health zones which we believe to have sufficient data for our analysis.

### Model variants

In the present study we present two previously developed models of gHAT, the first without animal transmission (denoted as "Model 4" in earlier work [16, 18, 19]) is a solely anthroponotic transmission model incorporating transmission between the tsetse vector and human populations [24]. In the model without animal transmission, a proportion of bites are assumed to be taken on animals, however non-human animals are assumed to be dead-end hosts. This formulation effectively includes two possibilities (i) that animals do not acquire the parasite and are therefore not infected, or (ii) that animals can become infected, however they cannot transmit the parasite back to tsetse—both of these result in identical tsetse-human infection

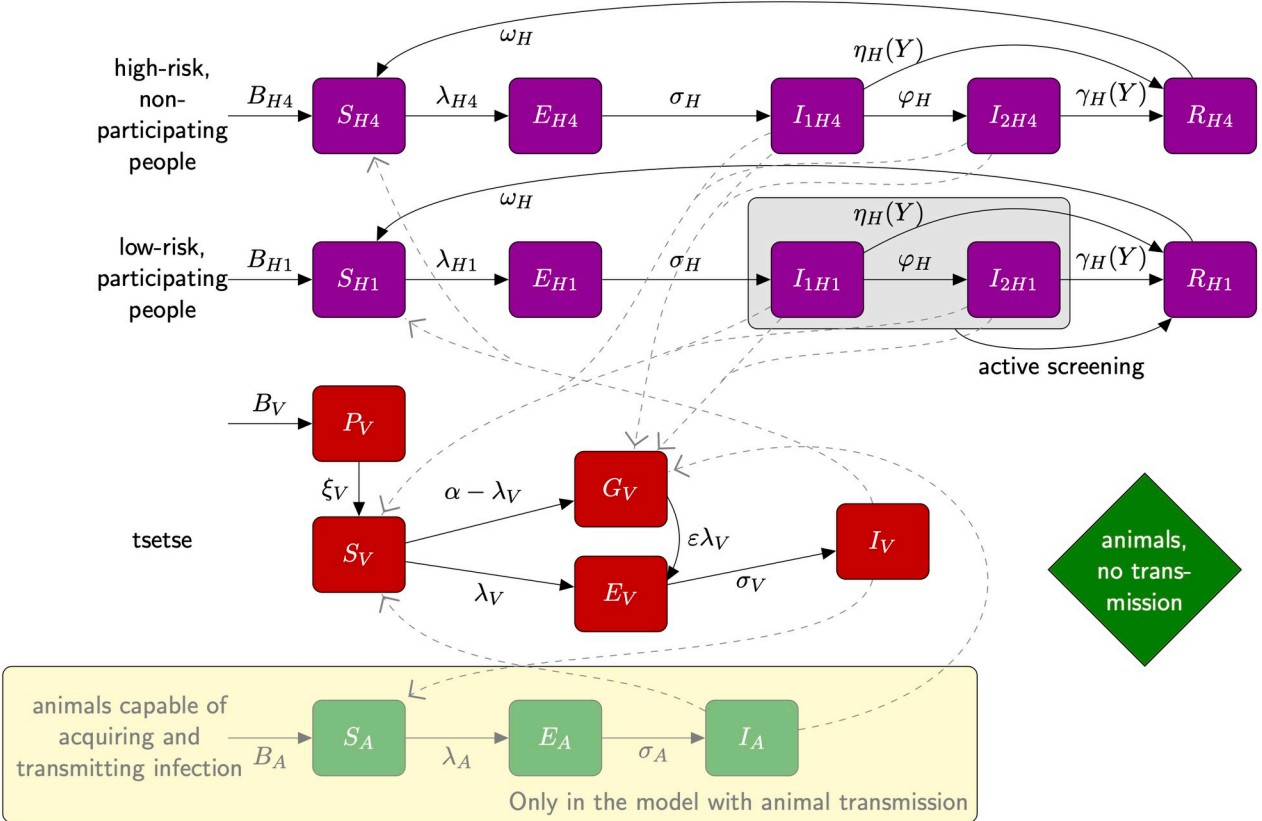

**Fig 1. Warwick gHAT compartmental model variants with and without animal transmission.** Purple boxes denote human infection states, with low-risk people who randomly participate in AS denoted with subscript *H*1 and high-risk people who do not participate denoted with subscript *H*4 (we retain this notation to align with previously published versions of this model). Tsetse are represented by red boxes and we explicitly include a pupal stage ($P_V$) and differentiate between unfed (teneral) flies ($S_V$) and non-infected but fed flies ($G_V$) to incorporate reduced tsetse susceptibility following the first bloodmeal. For simplicity, the exposed category for tsetse is shown as a single compartment, $E_V$; within the model this is subdivided into three compartments, $E_{1V}$, $E_{2V}$ and $E_{3V}$, to model parasite development. The pathway relating to the animal infection and transmission, which is specific to the model with animal transmission, is highlighted with a pale yellow box. Solid lines represent transition within a pathway, while broken lines represent transmission. More details and model equations can be found in S1 Text.

dynamics. In addition to a standard Ross-MacDonald formulation, there is also heterogeneity in the human population in terms of both risk of exposure to tsetse and participation in AS. This model structure has been fitted to a number of different data sets [16, 18, 19, 24, 25], although certain aspects of the fitting procedure and simulation of PS have evolved since it was originally published. In Rock et al. [16], Mahamat et al. [18], and Rock et al. [19], competing model structures were compared including those with possible participation of high-risk people in AS. In these studies the model with random participation of both groups was not statistically supported by the longitudinal data as it was not able to capture the observed trends as well as models with systematic non-participation of high-risk groups. This aligns with anthropological evidence [26] and programme experience which suggest certain groups (*e.g.* working age men) are often underrepresented in AS activities.

The second variant we consider is a model with animal transmission ("Model 7" in earlier work [16, 18, 19]). This model can be considered as an extension to the model without animal transmission, with the same risk and participation structure in humans. The model with animal transmission also allows for a proportion of bites to be taken on animals which can both acquire infection from, and transmit infection back to, tsetse. This animal reservoir may or

may not constitute a maintenance reservoir, and is in addition to the dead-end animal host population allowed for by the model without animal transmission. This formulation necessitates the addition of several extra model parameters: the effective animal reservoir density proportional to the human population, $k_A$; the proportion of blood meals taken on these animals, $f_A$; the mortality rate of an animal host, $\mu_A$, and the incubation rate of trypanosomes in animals, $\sigma_A$. Other model variants previously developed were not considered in this analysis as most have been shown to result in poor fits (without high/low risk structure and/or non-participation of high-risk people) [16, 18, 19]. Two of those models are extensions of the models included in this study but with all combinations of participation/non-participation in AS and risk possible. However these models are more complex (with more parameters) and had similar statistical support under model fitting in past studies. Therefore we focus here on the slightly simpler variants.

Aside from the host and risk-structure mentioned above, numerous other details about the progression of disease in humans and interventions are encoded in the formulation of both model variants (see S1 Text for full model equations). In brief:

1. Passive detection rates can increase over time so that infected individuals are detected and treated more rapidly now than in the past. We simulate this in all health zones of the DRC in 1998 when the CATT became available for general use. In addition, we simulate subsequent improvements in provinces in which there is evidence of a pronounced increase in the proportion of cases detected by PS found in stage 1 compared to stage 2; notably the former provinces of Bas Congo and Bandundu.

2. We assume that AS occurs at the beginning of the year and that only low-risk people participate (as supported by previous modelling studies [16, 18]) although high- and low-risk people have equal passive detection rates.

3. We account for imperfect diagnostics in the AS algorithm, with the standard screen-and-confirm process having a 91% sensitivity and very high but imperfect specificity (fitted independently for each health zone). In former Orientale province pre-2013, Médecins Sans Frontières (MSF) were known to be operating an AS algorithm based on CATT dilutions rather than parasitological confirmation. A positive CATT result at a dilution of 1:32 was declared to be a case and treated without parasitological confirmation. This is accounted for by modifying the parameters associated with the comparative algorithm for this period in this province: increasing the fixed sensitivity to 95%, and constraining the fitted specificity to be lower than that used outside this period. The use of video confirmation tools was introduced first in 2015 in Yasa Bonga and Mosango health zones in former Bandundu province and has subsequently been rolled out to the whole province. The microscopic parasitological confirmation procedure is video recorded and then reviewed centrally, to mitigate against false positive diagnoses. We simulate this as an increase to perfect specificity in 2018 for projections for other health zones in that province.

4. Whilst methods to control the tsetse vector are available, their wide-scale use was not implemented in the DRC during the data period with the exception of Yasa Bonga health zone in the former Bandundu province where Tiny Targets, which consist of a square of insecticide-impregnated netting alongside a square of fabric in a shade of blue found to be highly attractive to tsetse, have been used as a tsetse intervention since 2015 [12]. We simulate bi-annual deployment of targets in this region, and furthermore we simulate future strategies with and without possible Tiny Target-based vector control (VC) in other health zones. (See S1 Text for equations relating to VC).

## Fitting

Using a previously published Metropolis-Hastings MCMC algorithm, we fit each of both the models with and without animal transmission to the longitudinal human case data across the 158 health zones with sufficient gHAT data in the DRC. Crump et al. [24] provide more details on the algorithm and presents the fitting for the model without animal transmission. Of the four extra parameters required for the model with animal transmission we assume that $k_A$, the effective animal density proportional to the human population, and $f_A$, the proportion of blood meals taken on these animals, could be highly geographically heterogeneous and are therefore fitted within health zones. The other two parameters specific to the model with animal transmission, $\mu_A$, animal death rate, and $\sigma_A$, animal incubation rate, are held constant ($\mu_A = 0.0014$ days$^{-1}$ and $\sigma_A = 0.0833$ days$^{-1}$) across all health zones.

We use the same log-likelihood function as set out in Crump et al. [24] which matches model outputs of human case reporting to the longitudinal data for both active and passive detection modes, including staged reporting if this is available in the data. The full equations are found in S1 Text and include overdispersion in case detection to account for larger variance than expected under the binomial distribution. This log-likelihood function remains unchanged as no data on animal prevalence are available in the present study, however the two additional parameters fitted in the model with animal transmission are accounted for through their impact on the expected human case numbers.

**Priors.** Priors for all fitted parameters common to both models are given in S1 Text. For the effective animal density, $k_A$, a $\Gamma(1.26, 19.3)$ prior distribution was used; with a mode of 5.0, and 2.5% and 97.5% quantiles equal to 1.2 and 82.4, respectively. The proportion of blood meals taken on these animals, $0 \leq f_A \leq 1 - f_H$, where $f_H$ is the proportion of blood meals taken by tsetse on humans, had a flat/uniform prior over this range as any value seemed biologically plausible, and would be very dependent on tsetse feeding preference and animal abundance.

## Model comparison

Following the fitting of each model, model comparison was performed using Bayes factors (BF), which are ratios of the model evidence (or marginal likelihoods) under the two models. The BF may be expressed in terms of support for the model without animal transmission—with the model evidence from that model as the numerator, and the model evidence for the model with animal transmission as the denominator—or as its reciprocal if the evidence favours the model with animal transmission. For example, if for a particular health zone, BF for the model without animal transmission is 1.2, then the model without animal transmission is slightly favoured over the model with animal transmission, but the evidence is weak. Conversely if BF for the model with animal transmission is 102 then the model with animal transmission is decisively favoured over the model without animal transmission. Categorisations of BF value interpretations are given alongside results [27].

Importance sampled estimates of the model evidence for each model in each health zone were obtained following the methodology of Touloupou et al. [28], utilising a defense mixture [29]: in our study a weighted combination of a multivariate Gaussian mixture fitted to the 2 000 samples from the joint posterior distribution of the fitted model parameters (weight = 0.95), and the prior distributions of the fitted model parameters (weight = 0.05). See S1 Text for additional details.

## Host-specific reproduction numbers

For the model with animal transmission we can also assess the relative contribution of non-human animal reservoirs to transmission and assess their potential competence to maintain

infection in the absence of transmission in humans. Host specific, *i.e.* human or animal reproduction numbers were calculated using a next generation matrix approach [15] with a design matrix identifying contributions from the two host pathways. See S1 Text for full details.

## Projections

The results of the model fitting for the models with and without animal transmission, in the form of 1 000 realisations from the joint posterior distribution of all fitted parameters, were used to make projections under two main strategies. These are a subset of those reported in the paper on projections for the model without animal transmission [30]. In both strategies, AS was continued beyond 2016 at the mean level observed in that health zone in the period 2012 to 2016, this was chosen as reflecting an achievable level of AS in that locale with recent, if not current, resource availability. We present AS at higher coverage (the historical maximum in each health zone during 2000–2016) as a sensitivity analysis in our graphical user interface (GUI, online https://hatmepp.warwick.ac.uk/animalfitting/v1/), but do not focus on these results in the main manuscript. PS was assumed to continue at the 2016 level of effectiveness. The two strategies differ with regards to VC: in the basic strategy no VC was carried out, while in the second, VC was assumed to be implemented from 2020, with annual tsetse population reductions of 60%, which is a low, conservative estimate of potential VC effectiveness when this strategy is deployed at scale; this value is below tsetse reductions reported elsewhere in programme or study implementation which have reached up to 99% reduction in specific settings [12, 18, 31, 32]. Whilst we use this lower bound for VC effectiveness in the main manuscript, 80% and 90% tsetse reductions are presented as sensitivity analyses in our GUI to show how other reductions could impact transmission and reporting further. The VC exception is for Yasa Bonga health zone in the former Bandundu province, where VC was actually implemented from 2015. Note that in Yasa Bonga 90% tsetse reductions were reported [12] and assumed in the model after the first year.

Following the introduction of video confirmation of parasitology performed on all cases detected by AS across the former Bandundu province in 2018 we assume that the specificity of the AS algorithm is improved to 100% at this point and consequently there is a drop in case numbers detected by AS. This is particularly noticeable for our projections in Bokoro health zone.

EoT was assumed to have been reached when the number of new human infections went below 1 per year. The probability of EoT by any year was the proportion of projections in which EoT had been reached by that year.

The Matlab code used to perform the MCMC fitting to historical data, calculate host-specific reproduction numbers, approximate the model evidence and make projections is available from https://osf.io/3xadf/.

## Results

### Fitting and Bayes factors

We fitted both model variants to the health zone level data for 158 health zones. Fig 2 shows results for two example health zones, Bokoro in the former Bandundu province, and Tandala in the former Equateur province. Results for all other health zones can be viewed online in our companion GUI (https://hatmepp.warwick.ac.uk/animalfitting/v1/). Quantiles for the number of cases reported from AS and PS were of 10 000 stochastic samples, 10 samples for each of 1 000 random samples from the joint posterior distribution of the fitted model parameters. The estimated number of new human infections are taken from the solutions of the ordinary differential equations for the model for each of the 1 000 posterior parameter sets.

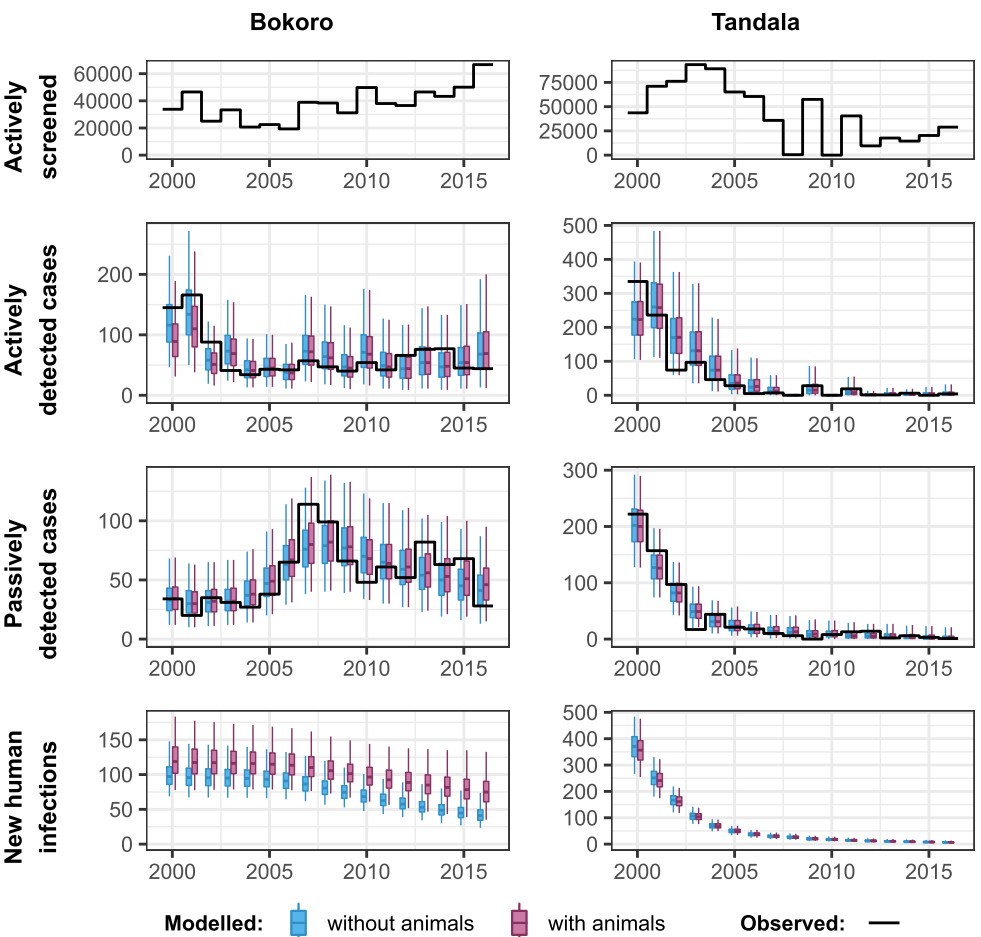

**Fig 2. Fits to historical case data for Bokoro and Tandala health zones, for models with (in pink) and without (in blue) animals contributing to gHAT transmission.** Reported data are shown as a solid black line and each boxplot shows the 2.5% and 97.5% quantiles of the model fit as the extreme point of the whiskers, 25% and 75% quantiles delimiting the box, and the median as the mid-line of the box.

The fit to the numbers of reported cases detected by both AS and PS is shown to capture the reported case trends well, including the observed 'humped' shape to the passive detection in Bokoro, which is also notable across other health zones of the former Bandundu province; this shape is associated with improvement to passive detection rates from both stage 1 and 2 of the disease. Both the models with and without animal transmission appear visually to have similar fits to the observed data points. However in Bokoro health zone, there is stronger statistical support for the model with animal transmission, including more new human infections each year and more unreported deaths when animal transmission is included (deaths not shown in the figure).

To compute the support for each of the model variants following fitting we categorised the BFs, where BF may be used to describe the statistical support for either model relative to the other, in a widely accepted way: Weak ('Barely worth mentioning'), $10^0 < BF < 10^{\frac{1}{2}}$; Substantial, $10^{\frac{1}{2}} < BF < 10^1$; Strong, $10^1 < BF < 10^{\frac{3}{2}}$; Very Strong, $10^{\frac{3}{2}} < BF < 10^2$; and Decisive, $BF > 10^2$ [27]. If BF for the model with animal transmission was greater than 1, the categorisation indicating statistical support for the model with animal transmission was used, otherwise

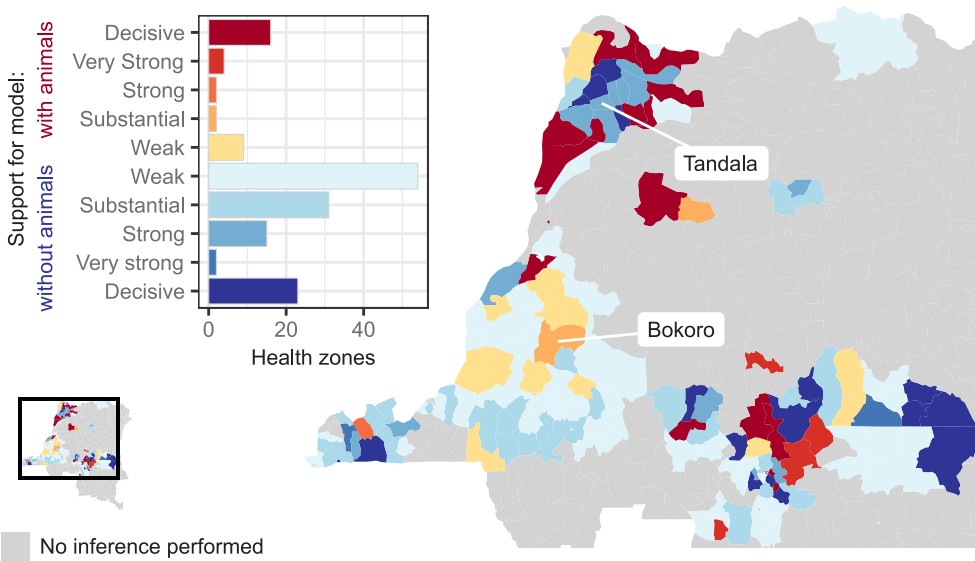

**Fig 3. Support for model either with or without animals contributing to transmission of gHAT.** Levels of support taken from whichever of the two Bayes Factors (BF, with either evidence for the model with or without animal transmission as the denominator) exceeded 1. Weak, $10^0 < BF < 10^{\frac{1}{2}}$; Substantial, $10^{\frac{1}{2}} < BF < 10^1$; Strong, $10^1 < BF < 10^{\frac{3}{2}}$; Very Strong, $10^{\frac{3}{2}} < BF < 10^2$; and Decisive, $BF > 10^2$. Health zones used as examples in later figures, Bokoro and Tandala, are indicated. Shapefiles used to produce this map were provided by Nicole Hoff and Cyrus Sinai under a CC-BY licence (current versions can be found at https://data.humdata.org/dataset/drc-health-data).

the categorisation used was for statistical support for the model without animal transmission. A map of our results is shown in Fig 3. The majority of health zones in which the model with animal transmission had substantial to decisive support (24 health zones in which the BF indicating statistical support for the model with animal transmission was greater than $10^{\frac{1}{2}}$) are in the former provinces of Equateur and Kasai Oriental (12 and 8 health zones, respectively). Many health zones had weak support for the model without animals.

Next, we used the posteriors from fitting the model with animal transmission to assess the host-specific reproduction numbers. Fig 4 shows the 2 000 estimates of the reproduction rate in humans ($R_{0_H}$) and animals ($R_{0_A}$) against one another for our two example health zones: Bokoro and Tandala. These posterior parameter samples can be summarised to give the probability that transmission in the animals only (where $R_{0_A} > 1$ and $R_{0_H} < 1$), in humans only ($R_{0_A} < 1$ and $R_{0_H} > 1$), in either humans or animals ($R_{0_A} > 1$ and $R_{0_H} > 1$), or in humans and animals ($R_{0_A} < 1$ and $R_{0_H} < 1$) would be capable of maintaining transmission of gHAT. There are no parameter sets in Bokoro or Tandala with a non-zero probability of animals being a maintenance host. In these health zones there is non-zero probability that humans could be a maintenance host for gHAT (0.03 in Bokoro, 0.70 in Tandala) or that transmission in both humans and animals is required to maintain gHAT transmission (0.97 in Bokoro, 0.3 in Tandala). These host-specific reproduction number results align with the substantial support for the model with animal transmission in Bokoro and strong support for the model without animal transmission in Tandala (see Fig 3).

We can do the same calculation for all health zones across the DRC. In 132 of the 158 health zones the most probable scenario was that transmission in both humans and animals would be required to maintain gHAT transmission. Fig 5 shows the results of the host-specific basic reproduction number calculations exemplified in Fig 4 summarised for all 158 analysed health

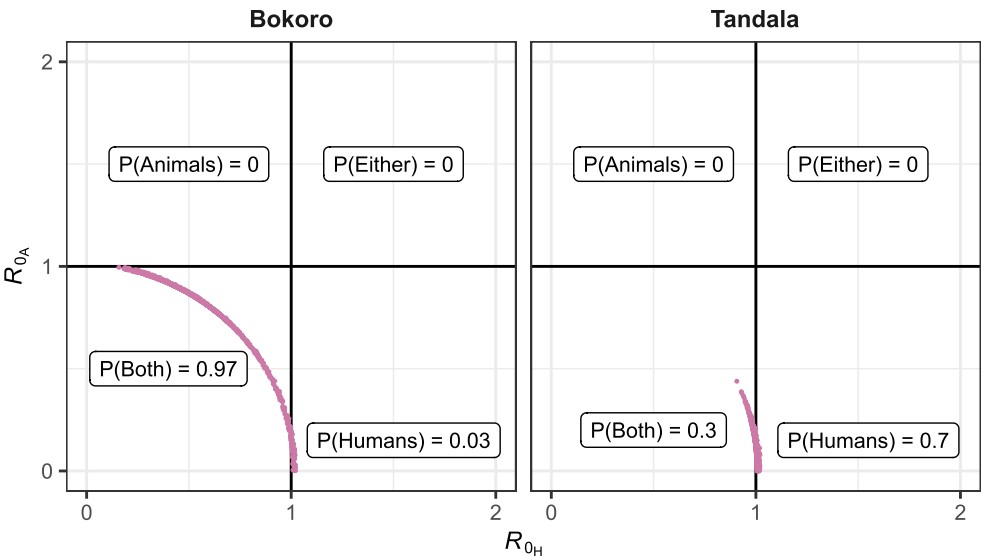

**Fig 4. Basic reproduction number for animals ($R_{0_A}$) and humans ($R_{0_H}$) for two example health zones, Bokoro and Tandala, calculated for each of 2 000 posterior parameter sets of the model including animal transmission.** The probability that transmission in animals only ($\mathbb{P}(\text{Animals}) = \mathbb{P}(R_{0_A} > 1, R_{0_H} < 1)$), humans only ($\mathbb{P}(\text{Humans}) = \mathbb{P}(R_{0_A} < 1, R_{0_H} > 1)$), both animals and humans ($\mathbb{P}(\text{Both}) = \mathbb{P}(R_{0_A} < 1, R_{0_H} < 1)$), or either animal or humans ($\mathbb{P}(\text{Either}) = \mathbb{P}(R_{0_A} > 1, R_{0_H} > 1)$) are sufficient to maintain on-going gHAT transmission are indicated.

zones. There were nine health zones in which some posterior samples resulted in $R_{0_A} > 1$, and hence non-zero probabilities that transmission in animals alone would be sufficient to maintain gHAT transmission. The highest value of this probability, 0.045, was observed in Lubunga 2 health zone in the former Kasai Occidental province. For Lubunga 2 health zone; the probability that both human and animal transmission are required for maintenance was 0.892, the probability that humans are a maintenance host was 0.063, and furthermore there was weak support for the model with animal transmission.

## Projections using models with and without animal transmission

We next used the posterior parameterisation of each health zone to predict future trends in case reporting and transmission. Bokoro and Tandala had averages of 24% and 7% annual AS coverage in the period 2012–2016, respectively. Fig 6 shows the projected impact of continuing AS at this level, alongside PS at the current level of effectiveness, from 2017 to 2040, on reported cases and human infections. With or without the presence of animal transmission, there are downward trends in these measures in these health zones. However there are more new cases detected by AS and PS reported each year, and more new human infections each year, under the model with animal transmission, with an impact on the achievement of EoT.

Our model results suggest that Bokoro will not reach EoT by 2030 under either model under the assumption that AS continues with the average annual number of people screened in the period 2012–2016 (48630 in Bokoro) and without VC (see Fig 6). Furthermore, if animal transmission is occurring, EoT is predicted not to be achieved for many years. However, the use of an intervention that targets transmission to and from both human and animal hosts should improve the situation. Fig 7 shows the impact that introducing VC in Bokoro from 2020 could have on the achievement of EoT in this health zone.

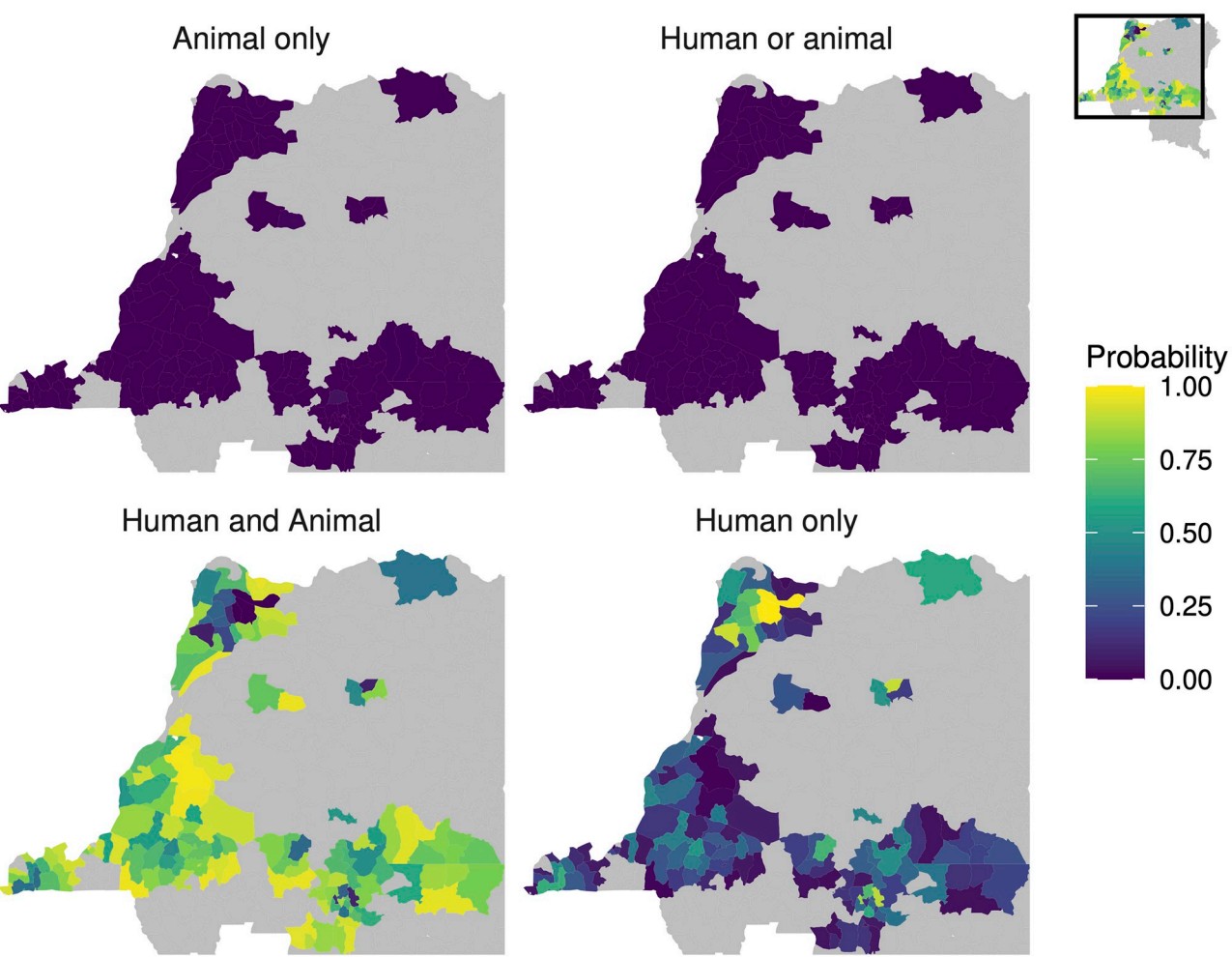

**Fig 5. Each subplot shows the probability the maintenance reservoir in each health zone of the DRC consists of: Just animals, either humans or animals, humans and animals combined or just humans.** In the bottom two subplots, humans are required to have sustained infection and animals could not maintain transmission of gHAT in the absence of human infection. Shapefiles used to produce these maps were provided by Nicole Hoff and Cyrus Sinai under a CC-BY licence (current versions can be found at https://data.humdata.org/dataset/drc-health-data).

The expected number of health zones that has achieved EoT in any year was obtained by summing each of the probabilities of having achieved EoT by that year. This was also considered for the case where VC was only carried out in health zones with less than 90% probability of having achieved EoT by 2030 without VC. These values are plotted as percentages of the 158 health zones analysed for both models and for the strategy with and without VC against year in Fig 8. An extended version of this figure is provided as Fig D in S1 Text including the results from fitting an ensemble of the two models weighted by the model evidence. The 90% probability cut-off removes 38 (under the model without animal transmission), 33 (under the model with animal transmission) and 36 (under the ensemble model) health zones from the group in which VC was carried out. As these health zones have a high probability of reaching EoT without VC, not performing VC in these locations has only a small impact on overall progress to EoT, particularly by 2030. It is clearly desirable, in terms of resources and costs to avoid such additional interventions where they are likely unnecessary.

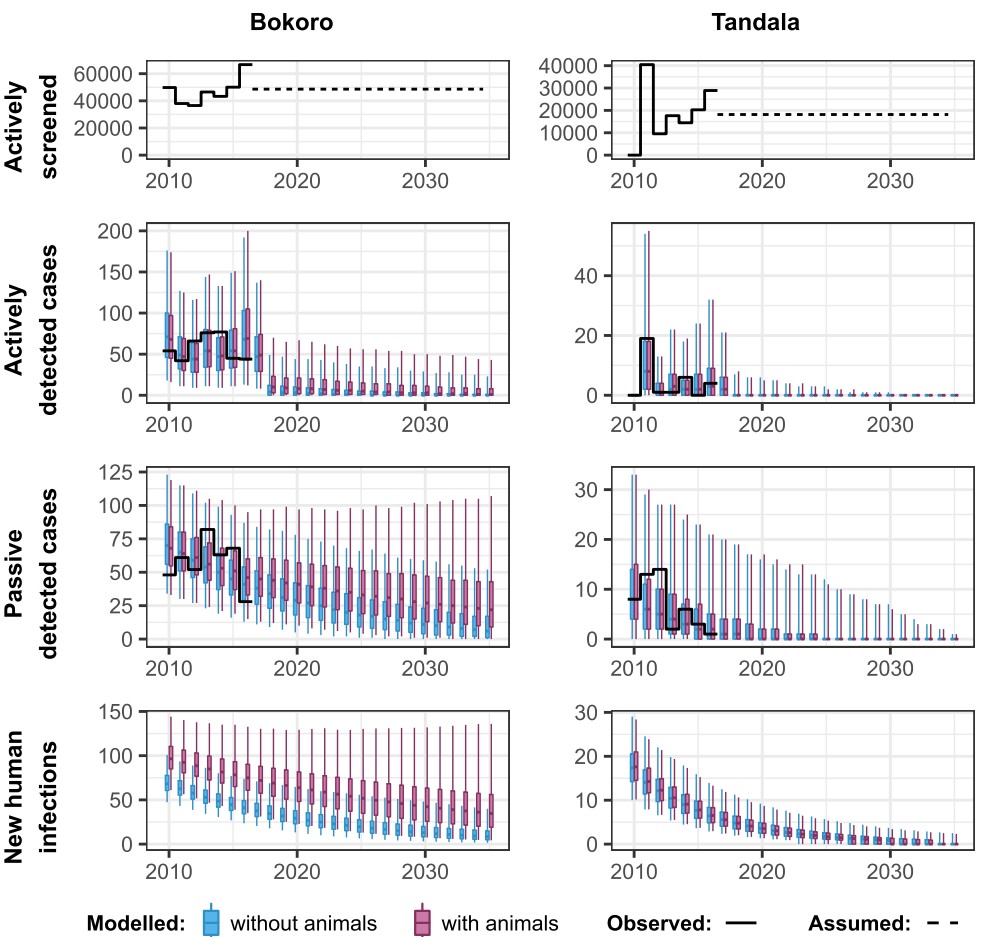

**Fig 6. Forward projections for Bokoro and Tandala health zone for models with and without animals contributing to gHAT transmission.** Within each health zone, future AS is assumed to be at the average level of screenings from 2012–2016, while PS continues at the 2016 level of effectiveness. In Bokoro, and other health zones in the former province of Bandundu, the specificity of AS was assumed to increase to 100% in 2018 due to video confirmation.

Results from both models and a model-evidence-based ensemble of the two are available online (https://hatmepp.warwick.ac.uk/animalfitting/v1/) for all 158 health zones analysed, including projection results for all strategies and posterior distributions of fitted parameters.

## Discussion

There are 24 health zones out of 158 in which there are substantial to decisive levels of statistical support for the model including animal transmission. In many of these health zones we identified a similar trend in the data that the health zones with support for the model with animal transmission are among those with a relatively long time series of low-level case detection by PS while also having been subject to reasonable AS coverage, if only historically. As case numbers fall, it is possible that if we were to repeat this study in a decade's time, there would be more support for the presence of an animal reservoir contributing to transmission of gHAT as more data become available. As with Guinea worm, signals from potential animal transmission should be most noticeable as we approach zero.

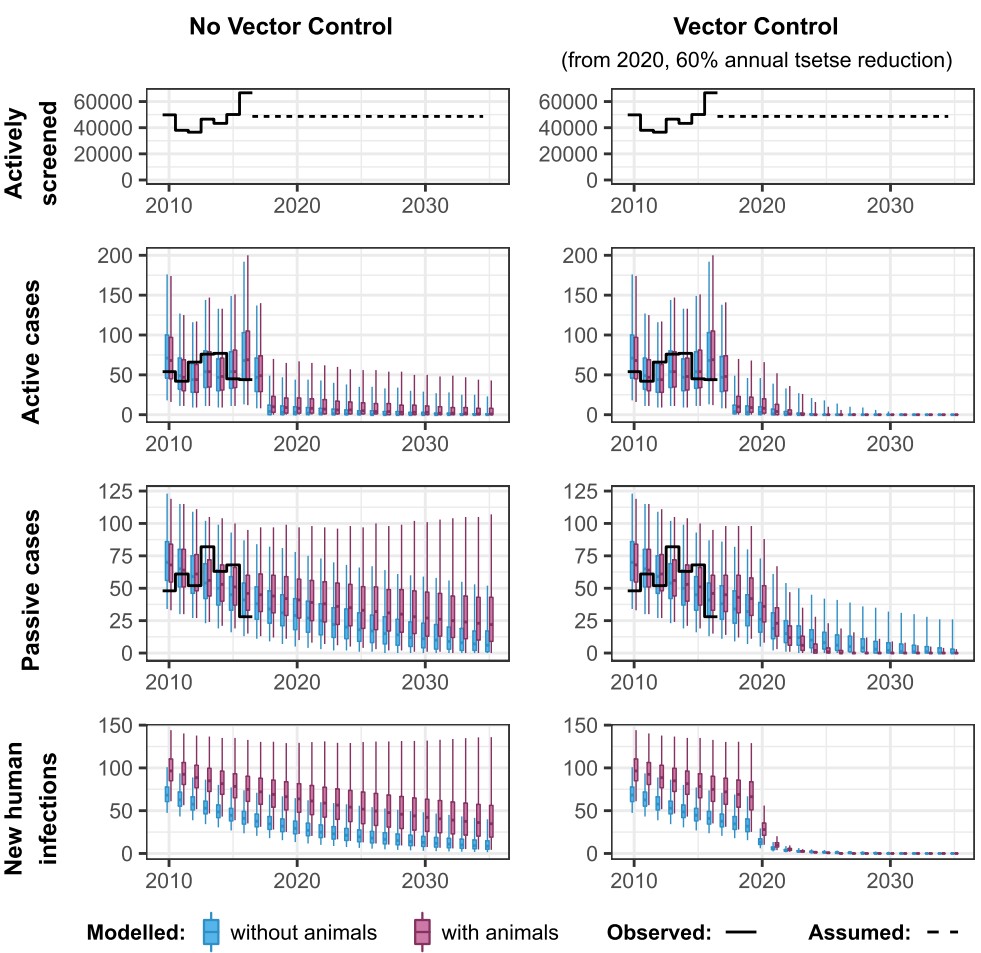

**Fig 7. Forward projections in Bokoro health zone for models with and without animals contributing to gHAT transmission and with or without Tiny Target-based vector control (VC) from 2020.** Future active screening (AS) is assumed to be at the average level of screenings from 2012–2016. In Bokoro, and other health zones in the former province of Bandundu, the specificity of AS was assumed to increase to 100% in 2018 due to video confirmation.

Some health zones with moderate case reporting in 2016 have weak to substantial support for animal infection. Kwamouth, Bagata, Mokala and Bokoro health zones—all located in the former Bandundu province—have had steady but relatively slow reductions in human gHAT reporting over the data period despite persistent, high-coverage AS. This could be an early (although inconclusive) signal that interventions may be being hindered.

More optimistically, this study indicates that it is extremely unlikely that an animal maintenance reservoir exists in the DRC—the highest estimated probability of this was 0.045, and only nine health zones had non-zero probabilities of this. In most health zones, if animals do contribute to transmission, the most probable scenario is that transmission in both humans and animals is required for maintenance of gHAT. Previous modelling in Cameroon, incorporating point estimates of animal prevalence, suggested that a combination of wild and domestic animals could constitute a maintenance reservoir [15]. The importance of our finding would be that interventions targeting transmission in humans will continue to reduce the prevalence of gHAT in the DRC and EoT should be reachable, but the existence of animal transmission may delay the achievement of EoT. The use of VC or other interventions that

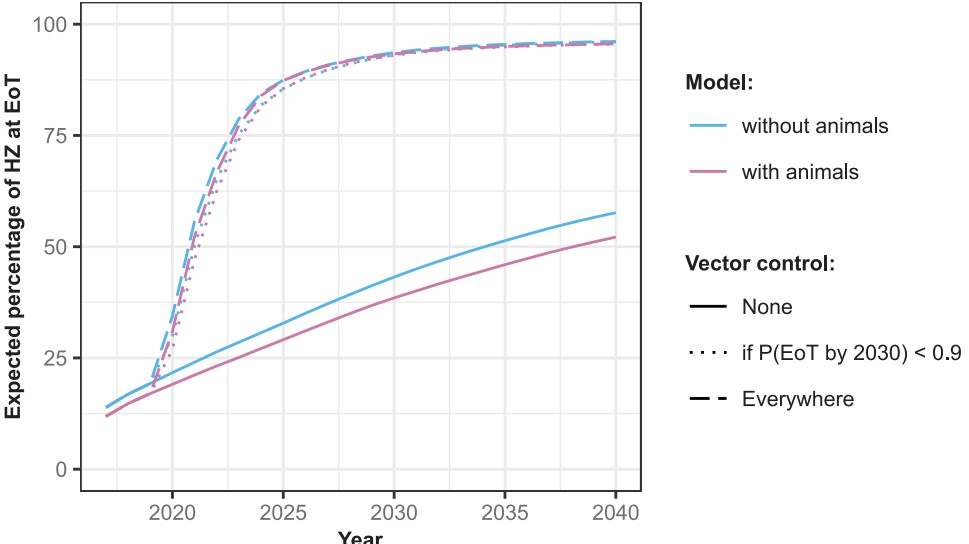

**Fig 8. The percentage of health zones studied expected to have reached elimination of transmission (EoT) against year.** Vector control (VC) was either simulated in none (solid lines), all (dashed lines), or a subset of health zones in which the probability of reaching EoT by 2030 without VC was less than 0.9 (dotted lines; using this cut-off measure, VC was simulated in 76% of health zones in the model without animal transmission and in 79% of health zones in the model with animal transmission).

impact transmission to and from all hosts may be highly beneficial, particularly in some regions.

Our ability to explain the levels of statistical support for either model studied here in terms of the historical data and in particular past interventions is constrained by the intimate connection between interventions—in particular the frequency and scale of AS—and case detections. This difficulty is exacerbated by the, unknown to us, impact of geographically variable levels of potential animal host species and habitat, tsetse population and habitat and proximity/overlap between all host and vector habitats. Two groups of health zones where we might have expected to find statistical support for the model with animal transmission should be considered. Firstly, those where interventions have brought about a reduction in case reporting to almost none and a relatively lengthy timeseries of low-level case reporting exists subsequent to this. This would be the case for health zones in Equateur, where high levels of AS activity in the early 2000s resulted in rapid reductions in the number of cases being reported followed by low-levels of case reporting detected by passive and AS in later years. Secondly, health zones such as Bokoro where the decline in case reporting is slower than might be expected given the on-going high levels of annual AS coverage.

In this study we have considered non-specific animal transmission, which may well consist of different species, different species' population sizes, and differing levels of interaction between each animal reservoir species, tsetse and humans across health zones. The purpose of this was to enable us to consider the potential importance of transmission to and from animals based only on the available human case data. In order to consider a specific animal reservoir species, considerable data collection would be required to quantify the epidemiology of gHAT within the species and transmission between the animal species, tsetse and humans. In addition to prevalence data, data on relative abundance of different animal species in relation to human or tsetse density, and feeding proportions of tsetse on animals are key pieces of information which would be required to understand the full picture of potential tsetse-animal

transmission cycles. If a suite of potential animal reservoir species were considered the recording effort would increase dramatically, and even then it would be appropriate to consider a catch-all non-specific animal reservoir as well to pick up all the species that we had neglected to study. It is very difficult to imagine a scenario in which sufficient data collection across time would allow resolution of differences between one or more specific potential animal reservoir species, and to date there has been very limited sampling of animal populations for gHAT globally [7].

Nevertheless, measures of location-specific prevalence of *T. b. gambiense* infection in animals could still provide valuable information for future modelling analysis, especially if very high prevalence or absence of infection is observed. At present there exists no 100% *gambiense*-specific test to identify conclusively that a non-human animal has *T. b. gambiense* infection rather than other animal trypanosomes, which are generally circulating at a much higher prevalence. Meanwhile tsetse blood meal analyses could help better parameterise models, narrow our uncertainty, and identify animal species of interest for sampling. We would recommend that data from the same locations at coinciding times for both animals and humans would best enable improved model fitting, and caution that the presence of non-specific *Trypanosoma*, or even *T. brucei* spp., unfortunately tells us little about transmission of the human-infective *T. b. gambiense* to and from animals. The present study could inform targeted animal sampling to collect animal data in either health zones with inconclusive evidence for or against animal transmission with the goal of improved certainty in particular where there is a large difference in projected EoT timing and the choice of model alters the strategy recommendation (see Fig B in S1 Text for a bivariate choropleth map of where high levels of uncertainty intersect with large difference in the probability of meeting EoT by 2030 between the models with and without animal transmission). As human prevalence is very low, there should be an expectation that large sample sizes would be needed to have moderate probability of finding animal infection where it does exist. The average within-year prevalence estimated from the model outcomes is presented in Fig E in S1 Text for humans, animals and tsetse in Bokoro and Tandala health zones, confirming that the prevalence in animals could be even lower than that found in humans. In Bokoro—where there is substantial statistical support for the animal transmission model—we estimated that at the end of the data period (2016) we would expect to have an infection prevalence of <0.1% in humans and a similar expected magnitude in animals. This value suggests that thousands of animal samples might have to be collected and analysed to have a moderate probability of finding possible animal infection with *T. b. gambiense*, and absence of infection in these samples would not rule out some animal transmission at even lower prevalences.

Between-location variation in factors that may be associated with animal transmission of gHAT; such as the animal species available, abundance of these species, tsetse density, tsetse feeding preferences and the size and proximity of the host and vector habitats, will all have an impact on the contribution of animals to transmission in a location. They would also be expected to impact on the ability of modelling of human case numbers to detect animal transmission in any given location. The success of human medical interventions would imply that any contribution of animals to transmission is not huge. However, the slow decline in cases in Bokoro health zone despite on-going high levels of coverage by AS (see Fig 2) may be indicative of a higher contribution. As evidenced by the very small number of cases being reported in, for example, Côte d'Ivoire, Uganda, Mali and Burkina Faso [13] the contribution of animals to transmission in these locations is likely low, if any. However, this may not indicate an inherent biological difference between the DRC and other countries but merely reflect that there is a greater variety in the animal/tsetse abundance and tsetse feeding preferences, which we also believe to be reflected in the range of results across the health zones of the DRC.

In the present study we have focussed on transmission models with high- and low-risk people with or without additional animal transmission. In our models we assume that some people may systematically not participate in AS activities but that they would be expected to either eventually be detected and treated through PS or die (possibly undetected) from gHAT disease. In regions where we find more statistical support for the model with animal transmission, it could be possible that alternative model variants could also generate similar patterns in observed case dynamics. One hypothesis is that undetected asymptomatic human infections with long-term asymptomatic carriage and ability to (potentially) self-cure could contribute to sustained transmission of gHAT [7, 33].

The model variants in the present paper already allow for long-term parasite carriage for a small proportion of individuals via their exponentially distributed infection periods, however not for self-cure or for skin-only parasite infection which would be missed in current screening algorithms. A few models considering potential self-curing asymptomatic transmission have been previously developed [23, 34, 35] and show that such a reservoir could hinder progress towards EoT. However, as with animal data, data on frequency and duration of such infections is sparse. Future modelling work should assess whether or not it is possible to distinguish signals in routinely reported gHAT case data which support animal transmission over asymptomatic transmission, or whether a mix of other types of human and/or non-human animal data would be needed to make this distinction.

The use of a deterministic model in this study, particularly in health zones where low prevalence has already been achieved, does raise questions around measurement of EoT and stochastic effects. A proxy threshold for our deterministic models is used here to estimate the year of EoT (less than one new infection per year to humans). From previous gHAT modelling using stochastic versions of the model without animal transmission, we know that, even at these very low prevalences, the deterministic and stochastic outputs match very well [36, 37]. Even though our underlying transmission model is deterministic, case reporting is generated by stochastic samples around fitted values and projections. The advantage of the deterministic approach is that it allows us to calculate a log-likelihood for each potential parameter set, which is far less computationally expensive than the approximations, such as in particle filtering MCMC or approximate Bayesian computation methods, required in fully stochastic model fitting. In the future there are a variety of potential options to further incorporate stochasticity—ranging from using posterior-specific proxy EoT thresholds [38], to stochastic projections from deterministic posterior parameterisation, through to fully stochastic model fits and projections. We do not expect that the use of any of these approaches would change our overall message presented here, but would be particularly valuable when answering questions such as how current data could indicate whether or not EoT has already been met, as has been done using the model without animal transmission [39], or surrounding the issue of resurgence following scaleback of intervention activities.

The geographical scale at which we have performed this study, the health zone level, is not an accurate representation of the distribution of either gHAT incidence or vector habitats. The model inherently assumes an equal mixing across the health zone. The presence of smaller pockets may lead to difficulty in achieving elimination but, when identified, also provide targeted sites for intervention activities.

The use of model evidence for model selection or comparison is regarded as the gold standard among Bayesian statisticians. The relative complexity of the models under consideration is automatically and naturally taken into account, without the use of an explicit penalty as is the case with other information criteria. An importance sampling technique was used here to approximate the model evidence using a parametric approximation to the observed joint posterior distribution of the model parameters [28]. This makes the use of Bayes factors feasible

and practical across the analysis of many health zones, as the calculation of the model evidence is a very small component of the overall analysis time (i.e. relative to the MCMC and projection of multiple intervention strategies).

In this study we have followed the Policy-Relevant Items for Reporting Models in Epidemiology of Neglected Tropical Diseases (PRIME-NTD) checklist [40] to ensure our results are transparent, convey uncertainty and have testable model outcomes. This checklist can be found in S2 Text.

## Conclusion

Health zones in which there was evidence in support of the model with animal transmission are concentrated in the former provinces of Equateur and Kasai Oriental. These health zones had low levels of on-going passive detection throughout the data period while AS activities also took place, particularly in the early 2000s. This may suggest that if similar patterns arise in other health zones—such that they reach low prevalence but not zero case reporting—support for the model with animal transmission may increase across the DRC. Despite this, the present analysis suggests that animals alone are not likely to be capable of maintaining transmission, rather the presence of animal transmission could slow down progress towards the EoT goal.

Under the model without animal transmission the expected number of health zones achieving EoT by 2030 under continuation of current intervention strategy is 43%, while it is 39% with animal transmission. If it were possible to implement VC at a large-scale across the DRC from 2020, with 60% annual reduction in tsetse population, then the predicted number of health zones reaching EoT by 2030 is 94% and 93% under models without and with animal transmission, respectively. This shows the benefit of targeting the tsetse as a means of preventing transmission to and from all hosts.

## Supporting information

**S1 Text. Additional methods and results.** More detailed description of materials and methods, and additional results and figures. **Fig B: Bivariate choropleth showing support for the models with or without animals contributing to transmission and the difference in the probability of achieving EoT to humans by 2030 from these two models**. Bivariate choropleth showing support for the models with or without animals contributing to transmission and the difference in the probability ($P_d$) of achieving EoT to humans by 2030 from these two models ("High" is more than 10% difference, "Medium" is 5-10% difference, and "Low" is less than 5% difference). Shapefiles used to produce these map were provided by Nicole Hoff and Cyrus Sinai under a CC-BY licence (current versions can be found at https://data.humdata. org/dataset/drc-health-data). **Fig D: The percentage of health zones studied expected to have reached elimination of transmission (EoT) against year, for the models without animal transmission, with animal transmission, and an ensemble of these two models**. The percentage of health zones studied expected to have reached elimination of transmission (EoT) against year, for the models without animal transmission, with animal transmission, and an ensemble of these two models. Active screening post-2016 was assumed to be at the mean level observed 2012–2016, and passive screening continued at the 2016 level of effectiveness. Vector control (VC) was either simulated in none (solid lines), all (dashed lines), or a subset of health zones in which the probability of reaching EoT by 2030 without VC was less than 0.9 (dotted lines; using this cut-off measure, VC was simulated in 77% of health zones in the ensemble model, 76% of health zones in the model without animal transmission, and in 79% of health zones in the model with animal transmission). **Fig E: Mean prevalence of infection within year as a percentage of the population**. Mean prevalence of infection within year

as a percentage of the population. Human cases are considered across disease stages and population risk group. In vectors, the prevalence of all infections and infectious (salivary gland) infections is presented.
(PDF)

**S2 Text. PRIME-NTD criteria.** Addressing the PRIME-NTD criteria for good modelling practises.
(PDF)

## Acknowledgments

The authors thank PNLTHA-DRC for original data collection, WHO for data access (in the framework of the WHO HAT Atlas [21, 22]), and Cyrus Sinai and Nicole Hoff from UCLA Fielding School of Public Health for providing health-zone-level shapefiles (current versions can be found at https://data.humdata.org/dataset/drc-health-data). We also thank the HAT MEPP scientific project manager Emily Crowley for her support in project organisation and proof-reading this manuscript.

## Author Contributions

**Conceptualization:** Simon E. F. Spencer, Kat S. Rock.

**Data curation:** Ronald E. Crump, Chansy Shampa, Erick Mwamba Miaka.

**Formal analysis:** Ronald E. Crump, Kat S. Rock.

**Funding acquisition:** Simon E. F. Spencer, Kat S. Rock.

**Investigation:** Ronald E. Crump.

**Methodology:** Simon E. F. Spencer, Kat S. Rock.

**Project administration:** Simon E. F. Spencer, Kat S. Rock.

**Software:** Ronald E. Crump, Ching-I Huang, Simon E. F. Spencer, Kat S. Rock.

**Supervision:** Simon E. F. Spencer, Kat S. Rock.

**Visualization:** Ronald E. Crump, Ching-I Huang, Paul E. Brown.

**Writing – original draft:** Ronald E. Crump, Ching-I Huang, Kat S. Rock.

**Writing – review & editing:** Ronald E. Crump, Simon E. F. Spencer, Chansy Shampa, Erick Mwamba Miaka, Kat S. Rock.

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
