## [Decision Letter · Decision Letter 0]

14 Feb 2022

Dear Dr Crump,

Thank you very much for submitting your manuscript "Modelling to infer the role of animals in gambiense human African trypanosomiasis transmission and elimination in DRC" for consideration at PLOS Neglected Tropical Diseases. As with all papers reviewed by the journal, your manuscript was reviewed by members of the editorial board and by several independent reviewers. In light of the reviews (below this email), we would like to invite the resubmission of a significantly-revised version that takes into account the reviewers' comments. 

We cannot make any decision about publication until we have seen the revised manuscript and your response to the reviewers' comments. Your revised manuscript is also likely to be sent to reviewers for further evaluation.

Sincerely,

Alberto Novaes Ramos Jr

Associate Editor

Anthony Papenfuss

Deputy Editor

Reviewer's Responses to Questions

**Key Review Criteria Required for Acceptance?**

**Methods**

-Are the objectives of the study clearly articulated with a clear testable hypothesis stated?

-Is the study design appropriate to address the stated objectives?

-Is the population clearly described and appropriate for the hypothesis being tested?

-Is the sample size sufficient to ensure adequate power to address the hypothesis being tested?

-Were correct statistical analysis used to support conclusions?

-Are there concerns about ethical or regulatory requirements being met?

Reviewer #1: The objectives of the study are clearly articulated with a clear testable hypothesis. I am not a mathematical modeller and am unable to comment critically on the modelling and statistical analyses; I presume that other reviewers will be able to do this. The models are based on previously published models produced by this research group and so on that basis I assume that the models etc are appropriate and correctly implemented.

Reviewer #2: Gambiense human African trypanosomiasis (gHAT) is still a major public health burden in the Democratic Republic of Congo. Although Trypanosoma brucei gambiense is believed to be transmitted primarily between humans due to Tsetse fly bits, the role of an animal reservoir in maintaining disease transmission is unclear. 

The goal of this study was to look for evidence of gHAT transmission from an animal reservoir and consider how the existence of such reservoirs might effect efforts to eliminate transmission of gHAT in the DRC. 

The main methods of this report were (1) formulation of deterministic models of gHAT transmission via the Tsetse fly, and (2) providing statistical support (in a Bayesian formalism) for the most likely models based on the actual public health data from the health zones of the DRC. Multiple models are constructed for individual health zones (choosing many parameters from a statistical distribution). The authors then calculated the Bayes support of models without animal transmission to models with animal transmission. The models employed are very similar to ones used previously by the authors, but in this report the authors are concerned in gauging the importance of possible animal reservoirs in blocking or slowing efforts to end transmission of gHAT in the DRC. 

The model calculations suggests that in some health zones, animals do indeed serves as a reservoir of Trypanosoma brucei gambiense. In those zones, their models suggest that even a modest vector control effort wouldl contribute greatly to the goal of elimination of transmission of gHAT.

One could quibble about the details of specific parts of the human/tsetse/Trypanosoma system in their modeling schema, but the development of the models from real data and their evaluation and statistical weighing are impressive and thorough. The authors recognize in the final Discussion that non-deterministic models might be better in capturing the effects of stochastic fluctuations, but they argue that from previous experience, deterministic and stochastic models yield similar results, and that the use of fully stochastic models is unlikely to change the major conclusions that Tsetse control is needed to reduce transmission from local animals to humans.

Reviewer #3: The objectives are clearly articulated with a clear testable hypothesis, the design, population and analyses were appropriate, sample size sufficient, and ethical concerns were met.

Major Question 1: In the models, why does active screening only capture low-risk people? Maybe give a bit more of background on why the previous studies found this. 

Page 5 Point 3: How do CATT dilutions and video methods work to ID gHAT?

Page 5 Point 4: What is a Tiny Target?

Page 5 Model Variants: It might be good to explain why the other models (1-3, 5-6, anything after 7?) were not selected?

Reviewer #4: ### Major comments

Figure 1 Caption (and text elsewhere) suggests that model equations can be found in the supp, but I couldn't find them unless they meant Table S1.6 and S1.7. Can the equations or equivalently a fully annotated model diagram (i.e. with the rates of all state transitions) be provided in the supplementary materials please?

S1.2 Birth rate: Can the authors justify using the fixed population size assumption? My understanding is that the population of DRC is growing rapidly (possibly >60% over the period covered by the data) and will also probably grow until 2040. I could imagine this could have some significant impacts on the data fitting for passive cases where the denominator is the *whole* population (with access to healthcare)

S1.2 Overdispersion for surveillance is fixed without a reference? How were these values chosen?

S1.5: What is an active/passive case (denoted $A_{M1}$ and $P_{M1}$)? How do these related to the compartments in Figure 1?

How are the models initialised?

The definition of EoT for the purposes of classifying model projections (less than 1 new case per year) needs to be stated earlier (I think this is first mentioned towards the end of the discussion). However the discussion of the limitations of using an ODE model in this way is handled very well!

### Minor comments

Line 111: Why was 13 datapoints chosen as the cutoff? Probably not worth a sensitivity analysis, but might be worth knowing why this cutoff was chosen. The text explains why it is more than 10 (in previous studies), but this doesn't really explain the choice of number.

Line 116-178: Seems out of place, i.e. should be in the limitation section of the discussion. I would expect the data subsection of the methods to describe the data that was used, with minimal commentary.

Implicitly assumes that populations are well-mixed within districts. Smaller pockets can lead to difficulties in elimination efforts. This is a very reasonable assumption given the sparsity of data, but could be mentioned as a limitation. Might also be a alternative explanation to passive surveillance declines being slower than active surveillance

S1.5.1 The way the likelihoods for stage 1 and 2 cases are split up are a little odd and took me a while to decide whether they make sense. A seperate BetaBin model for the observed number of $A_1$, $A_2$, $P_1$, $P_2$ cases would seem more natural to me. I'm guessing they formulated it this way because they didn't have the split by stage 1/stage 2 cases at every sample point and would simply drop the Bin model terms when this is not available. This section deserves a *little* more explanation. 

S1.4: Please provide details of how tsetse fly reduction is modelled here. This section was not really intelligible to me. Though I think it's fine to leave out *details* and refer readers to previous work, I believe that what's written in the paper and supp should be *intelligible* on its own. For instance, what does 'probability of both hitting a target and dying' mean? (After re-reading I'm guessing this refers to a tsetse fly hitting a lure/trap/device and dying, but at first I thought this was talking about public health elimination targets and was very confused!) What are $f_T$, $t$, and $f_{max}$? What is 'target efficacy' and why is it only around 0.03 and not .60 in the figures? Why does a 60%/90% tsetse reduction reduce tsetse numbers by more than 60%/90% in the figures (looks more like 75%/95% from the figures?) Is this to do with the timeframe over which the interventions are applied? If in the *long-term* tsetse reductions will be much more than the stated 60%/90% then does it need to described differently (e.g. long-term average psuedo-equilibrium tsetse abundance reduce by X%)

S1.5.2 It's not clear what this imputation achieves, if anything. Surely imputing the failures with a NegBin to then use in the denominator of a Bin, nets no useful information for estimating parameters of interest $\\theta$? This is perhaps what they mean by 'canceling from the posterior', but I'm not sure what they are trying to say here. If it just drops out, why not just omit the data point? I'm familiar with imputation techniques for regression studies where there a few *covariates* missing and people don't want to throw away all the other non-missing covariate information; in these methods the approach is to use some information about patterns/associations within covariates to infer what the missing covariate might be. The (weak) analogy for the current study would be to use data on active surveillance denominators from other previous surveys in the focus (or other foci) to inform what the probable denominator might be for the focus with missing denominator. Though in principle I would expect this approach to provide some additional information about $\\theta$, I'm not suggesting they have to do something like this since it seems a lot of effort for not much gain. Also calling it a hierarchical prior is a bit odd! I would have thought that $A_D^{-}|\\theta \\sim NB(A_D(t), P^{+}(t))$ is the model and that (implicitly) there is only an improper uniform discrete prior for $A_D^{-}$ over the integers here. Keeping the above as their model and adding an actual informative/empirical prior on $A_D^{-}$ seems like a natural way to net more information on $\\theta$. I could have probably said this all more succinctly, but in summary, what the authors have done is probably harmless, but it might have some odd implications for the fitting if the imputation doesn't simply 'cancel out'.

**Results**

-Does the analysis presented match the analysis plan?

-Are the results clearly and completely presented?

-Are the figures (Tables, Images) of sufficient quality for clarity?

Reviewer #1: The results match the analysis plan and are generally clearly presented.

I found the figures difficult to read - the colour scheme of the boxplots in figures 2 and 4-7 are particularly difficult.

Reviewer #2: Yes, the analysis presented matches the planned analysis. The figures in this paper and in the two supplemental files are well done and indicate both the geographic distribution of gHAT in the DRC health zones, and the results of their models. Figures 2 and 6 were especially informative in helping to visualize the implications of their model results.

Reviewer #3: The analysis matches the plan, results are clearly and completely presented, figures are clear.

Major Question 2: The authors find that animal involvement in maintaining transmission cycles is supported in Bokoro but not in Tandala. I noticed the two health zones have very different distributions of active testing, and I didn’t see much discussion of this. Did the authors explore how the testing strategy/frequency/intensity could impact the results? Apologies if I missed this in the details somewhere.

Reviewer #4: The results here match the methods and provide a mostly clear picture of the results. There a number of minor presentation points which I have included in a later section. However, a couple points of more considerable concern are listed here:

Lines 277-279: "Both Model 4 and Model 7 fit very similarly to the observed data points. However in Bokoro health zone, which has strong support for the model with animal transmission…" These two statements appear rather contradictory. Please clarify

Fig 7: The sharp drop-off (2017-2018) then slow decline (2019-2040) of predicted actively identified cases warrants explanation. Almost looks like a bug!

**Conclusions**

-Are the conclusions supported by the data presented?

-Are the limitations of analysis clearly described?

-Do the authors discuss how these data can be helpful to advance our understanding of the topic under study?

-Is public health relevance addressed?

Reviewer #1: The conclusions are supported by the data presented but I felt there were disparities in tone between the Abstract/Author summary and the Discussion. In particular:-

The Abstract reports moderate to high evidence of statistical support for some animal transmission.

Author summary reports 'statistical evidence' for animal transmission - ie nothing to qualify the strength of evidence

Discussion reports 24 health sones with 'substantial to decisive levels of statistical support'.

I think a consistent and conservative ('substantial to decisive') conclusion about the role of animals would be welcome across all sections and, if space allows, some indication of how the 'substantial to decisive' conclusion is made.

I also found that important conclusions in the Discussion did not find their way to the Abstract. For instance, the conclusion that 'it is extremely unlikely that a maintenance animal reservoir exists in DRC' is an important finding for policy makers and therefore should be in the Abstract. My understanding is that the authors conclude that animal reservoirs are not preventing elimination of transmission but may delay its achievement.

I think the Discussion could merit some more discussion of the relevant literature and the practical implications of their work. For instance, some of the conclusions that I think might emerge from such discussion are:-

It is practically impossible to sample sufficient number of potential animal hosts to test the hypothesis that animals are acting as reservoirs. This seems to be the implication of the sentence the 'prevalence in animals is even lower than that found in humans'. If prevelance in humans is approaching 1/10,000 then presumably the predicted prevalence in animals is <1/10,000. Expanding on this would guide researchers and funders concerned with identifying animal reservoirs empirically.

The stronger evidence for animals being reservoir hosts was related to certain health zones which, I think, were also places with low case incidence following intensive active screening. The authors speculate that this is likely to occur in other health zones as they move to this low incidence scenario. Do the authors think that this the role of animal reservoirs is obscured in high incidence settings - rather like guinea worm - or is this an artefact of the models? Are there other hypotheses to explain the difference between Health Zones? For instance, do parts of Equateur have higher densities of livestock?

Several countries appear to have achieved or be approaching the elimination of transmission (e.g. Mali, Burkina Faso, Cote d'Ivoire, Uganda). Does this fit with the notion that there are animal reservoir hosts for gHAT. Do the authors think that DRC is different? Could this be a contribution to the high disease burden?

Some discussion about published empirical evidence for animal reservoirs being important would strengthen the discussion.

In the Abstract, I found the use of numbers of health zones with evidence of some animal transmission (24/158) followed by EoT being reduced from 68 to 61 Health Zones if animals are reservoir hosts, and then back to 147/158 achieving elimination with the addition of vector control rather confusing. I thing it would help to use the same denominator (158) so the reader can judge the importance of the finding and proposed solution.

Reviewer #2: The data presented are the results of their modeling, and do support their conclusions that (1) some of the health zones in the DRC do have gHAT reservoirs and (2) some effort is needed to control the Tsetse fly vector in those health zones in order to reduce gHAT transmission. These conclusions are of the upmost importance in guiding efforts to control or eliminate gHAT in the DRC.

Reviewer #3: Conclusions supported by the data, limitations clearly described, discussion of insight provided was great, and public health relevance was clear.

Reviewer #4: I believe the conclusions are mostly very sound and that most of the main limitations are clearly stated with one main exception.

Throughout the paper, the paradigm is that of three kinds of vertebrate hosts, 'low risk humans (actively and passively screened)', 'high risk humans (only passively screened)' and 'other animals'. However, it appears to me that a model that replaces the 'other animals' with a third class of humans who are not actively or passively screened might have a similar model form and very similar model predictions/likelihood (the main difference would be the life expectancy of the hosts, but I'd be surprised if this makes much difference to the model fit). This is not a problem in the construction of the models per se (they appear to have done an excellent job of working with the available data!), but I think it means that the results need to be interpreted a little more cautiously. For instance, health districts with 'strong evidence' for the animal models might equally be districts with a human subpopulation that isn't captured (or very poorly captured) by active/passive surveillance (e.g. certain groups defined by access to healthcare, geography, age, sex, disability, social standing, individual behaviour etc.). This probably doesn't affect the predictions about time to EoT and impact of tsetse reductions, but looking for an animal host might prove a wild goose chase! I'm not suggesting they need to fit or consider and alternative model, but something along these lines might need to be raised in the discussion.

**Editorial and Data Presentation Modifications?**

Reviewer #1: The article is well produced and written.

Reviewer #2: The only criticism I have is that the authors state in the "Data Availability" statement that "No - some restrictions will apply". They then state "The gambiense human African trypanosomiasis (HAT) data were obtained from the WHO HAT Atlas and are subject to a data sharing agreement. Interested parties should apply to the WHO in order to gain access to these data."

I have some qualms about this restriction, especially as the credibility of the WHO has come under attack because of the fraught relationship between the WHO and China over the origins of SARS-CoV-2. I state upfront that I think those attacks are mostly unfair to WHO, but then, in the case of gHAT, what is the criteria for an "interested party" to gain access to the data? Is patient confidentiality the main reason for the restriction? It could be that PLoS and the authors have already negotiated an agreement about the restricted access based on legal and ethical concerns, but I have no information about that.

Reviewer #3: Page 2 line 29: realized —> “achieved” is a better word here.

Page 2 line 29: also, “historically assumed” - is there a reference for this? Do the authors mean the models have only ever just considered human-to-human transmission?

Page 2 line 39: directly are —> directly is

Page 3 line 58-59: cite these studies here, either after this clause, or at the end of the sentence.

Page 3 line 66: already defined DRC, so just use DRC.

Page 3 line 79: build upon this prior work, using…

Page 5 line 188: assume, not believe

Page 7 line 242: great tool!

Page 8 Figure 3 caption: why specify “ ‘ Barely worth mentioning’ “? Just say “Weak”. Let the reader decide the worth of that result. 

Page 11 Line 394 - yes!!!

Reviewer #4: Line 131 and 143 (and elsewhere): Reusing Model numbers from previous work is confusing. If the name being reused was informative, then reuse would be fine. I think it would be clearer to give them informative names (for instance something like: Model 4 -> anthroponotic model) or new uninformative names (Model A, Model B…). Then at key points they could link back to previous work (e.g. "Model X is equivalent to Model 4 from previous work")

Figure 1: Green diamond say non-reservoir animals. Should this be non-human animals? The question is whether or not they are reservoirs, no?

Figure 1: No dotted (transmission) lines to or from $H_4$ compartments (though from Table S1.6 I gather there is transmission to/from $H_4$.

Figure 1: Adding legend/text describing what the different lines (dotted/solid) mean which might help non-modelling readers.

Line 156 (and elsewhere): 'Passive detection rates'. I assume here the authors mean the probability of detection through passive surveillance and not simply the incidence of cases reported by passive detection? Some authors insist that the word 'rate' be reserved to talk the frequency of events per unit time (e.g. death rate due to cancer would mean the number of people dying due to cancer per year/month/etc., not the proportion of cancer patients who die from the disease) . I think this is good practice in general (though I try not to be pedantic) but using the term rate here (and elsewhere?) is genuinely confusing! There is also a missing 'and' or fullstop in line 156.

S1.6: Side by side, $\\mathbb{1}^n$ and $\\mathbb{0}^n$ suggest blocks of ones and zeros respectively which is confusing. I'm guessing the $\\mathbb{1}^n$ in the equations are meant to be identity matrices (usually notated $I_n$ ) rather than matrices of all ones (usually notated $J_n$)? Also this whole section might be more easily summarised in terms of sub-matrices of $K$.

Fig 2 and 6: The y-axis labels need to be clarified. After lots of reading between the lines I think they mean (from top to bottom) Number of people actively screened per year, Number of cases reported from active screening per year, number of cases reported from passive screening per year and number of new infections per year. I think the confusion was because 'active cases' usually means 'the number of people who are known to infected right now'. Perhaps 'actively detected cases' and 'passively detected cases' would clarify this. A similar change throughout the paper might be helpful. The cases themselves are not 'active' or 'passive'!

Fig 2 and 6: It might also be helpful to indicate the populations of the two health districts to get a sense of proportion.

Line 329: What is the MeanAS strategy?

Line 61/62: I had to re-read this a few times — I think the paragraph break threw me off by making it unclear what 'these studies' (line 62) referred to.

Line 72: Perhaps better if they use the phrase 'maintenance reservoirs' here and throughout. It's introduced above and used again to state the goals of the study

Lines 82-84: 'If there is animal transmission' — this qualifier is unnecessary. They are looking at trends for both kinds of models. Perhaps they meant something like: Based on current trends, what will projections to 2030 and beyond look like and what role does animal transmission play in this (if any)?

Fig 3: The caption is overly technical with the regards to Bayes factors. The description doesn't help people who don't understand Bayes Factors or people who understand Bayes Factors (who just need to know the cutoffs). I suggest dropping everything in parentheses and replace all the $K_{ij}$ with 'BF' (or some other shortening for Bayes factor)

Line 323: phrase 'at this level' is redundant?

Line 360: 'observed data timeseries grows' sounds like they are talking about growing incidence, though I'm guessing they mean 'as more data becomes available'

**Summary and General Comments**

Reviewer #1: (No Response)

Reviewer #2: As stated above, the efforts to develop realistic models of gHAT transmission and provide good statistical support for them are impressive and thorough. The authors are concerned about using their results to guide real-world efforts to control gHAT in a country plagued by this disease, and I think this paper gives good support for the need to control the Tsetse fly vector. 

This reviewer wishes that the authors could have had space to present some stochastic models. But perhaps that needs a more theoretical paper than this one, which is clearly aimed at influencing the implementation of needed public health measures. As the discussion and results stand now, this manuscript is complete (aside from the data access issue mentioned in the Editorial and Data Presentation Modifications section).

Reviewer #3: The current article “Modelling to infer the role of animals in gambiense human African trypanosomiasis transmission and elimination in DRC” by Ronald E Crump and colleagues is a well-written and important contribution to the fight against an NTD that can circulate in both human and non-human hosts. This sort of one-health modeling approach is necessary for a realistic estimation of the capacity of EoT programs to achieve their goals. The methods appear rigorous, claims are supported by the results shown, and this study makes a valuable contribution to the field with clear implications for public health.

I have some questions for the authors, though. See above.

Reviewer #4: This is in my opinion a very interesting and well executed study. The authors have done an excellent job of working with the available data. The introduction lays out the complex possible relationships that could exist between multiple reservoirs and the rest of the paper continues with this clarity of purpose and outcomes. Their use of interactive data visualisation and maps is excellent. Though I have identified an unfortunately long list of questions and issues, they are all individually pretty minor and mostly around presentation and interpretation. I do not think any further experiments are necessarily needed (unless the odd outcomes in Figure 7 turn out to be a real bug!). I have no concerns about dual publication, research ethics, or publication ethics

PLOS authors have the option to publish the peer review history of their article (what does this mean?). If published, this will include your full peer review and any attached files.

Reviewer #1: No

Reviewer #2: Yes: Philip G McQueen

Reviewer #3: No

Reviewer #4: Yes: Angus McLure
---

## [Decision Letter · Decision Letter 1]

22 Jun 2022

Dear Dr Rock,

We are pleased to inform you that your manuscript 'Modelling to infer the role of animals in gambiense human African trypanosomiasis transmission and elimination in DRC' has been provisionally accepted for publication in PLOS Neglected Tropical Diseases.

Best regards,

Alberto Novaes Ramos Jr

Associate Editor

Anthony Papenfuss

Deputy Editor

Reviewer's Responses to Questions

**Key Review Criteria Required for Acceptance?**

**Methods**

-Are the objectives of the study clearly articulated with a clear testable hypothesis stated?

-Is the study design appropriate to address the stated objectives?

-Is the population clearly described and appropriate for the hypothesis being tested?

-Is the sample size sufficient to ensure adequate power to address the hypothesis being tested?

-Were correct statistical analysis used to support conclusions?

-Are there concerns about ethical or regulatory requirements being met?

Reviewer #1: The authors have responded strongly to all my comments and questions.

Reviewer #4: (No Response)

**Results**

-Does the analysis presented match the analysis plan?

-Are the results clearly and completely presented?

-Are the figures (Tables, Images) of sufficient quality for clarity?

Reviewer #1: The authors have responded strongly to all my comments and questions.

Reviewer #4: (No Response)

**Conclusions**

-Are the conclusions supported by the data presented?

-Are the limitations of analysis clearly described?

-Do the authors discuss how these data can be helpful to advance our understanding of the topic under study?

-Is public health relevance addressed?

Reviewer #1: The authors have responded strongly to all my comments and questions.

Reviewer #4: (No Response)

**Editorial and Data Presentation Modifications?**

Reviewer #1: The authors have responded strongly to all my comments and questions.

Reviewer #4: (No Response)

**Summary and General Comments**

Reviewer #1: This is an important paper which should be accepted for publication in PLoS Neglected Tropical Diseases

Reviewer #4: As I said in my previous review I believe this is a well-executed and sound study. The reviewer response letter and the updated manuscript demonstrate attention to detail. My previous comments have all been addressed or responded to well. I have no outstanding concerns.

PLOS authors have the option to publish the peer review history of their article (what does this mean?). If published, this will include your full peer review and any attached files.

Reviewer #1: No

Reviewer #4: **Yes: **Angus McLure

---

## [Editor Report · Acceptance letter]

7 Jul 2022

Dear Dr Rock,

We are delighted to inform you that your manuscript, "Modelling to infer the role of animals in *gambiense* human African trypanosomiasis transmission and elimination in the DRC," has been formally accepted for publication in PLOS Neglected Tropical Diseases.

Best regards,

Shaden Kamhawi

co-Editor-in-Chief

Paul Brindley

co-Editor-in-Chief
